# Olfactory recovery following infection with COVID-19: A systematic review

**Ali Jafar** [1], **Andrea Lasso** [2], **Risa Shorr** [3], **Brian Hutton** [4‡], **Shaun Kilty** [3,5‡] *

**1** Department of Otolaryngology - Head and Neck Surgery, University of Ottawa, Ottawa, Canada, **2** Ottawa Hospital Research Institute (OHRI), Ottawa, Ontario, Canada, **3** The Ottawa Hospital, Ottawa, Ontario, Canada, **4** Knowledge Synthesis Unit, Ottawa Hospital Research Institute (OHRI), School of Epidemiology, Public Health and Preventative Medicine, University of Ottawa, Ottawa, Ontario, Canada, **5** Department of Otolaryngology - Head and Neck Surgery, University of Ottawa, Ottawa Hospital Research Institute (OHRI), Ottawa, Ontario, Canada

☺ These authors contributed equally to this work.
‡ BH and SK also contributed equally to this work
* skilty@toh.ca

**Data Availability Statement:** All relevant data are within the manuscript and its Supporting information files.

**Funding:** The authors received no specific funding for this work.

## Abstract

Olfactory loss has been identified as one of the common symptoms related to COVID-19 infection. Although olfactory loss is recognized, our understanding of both the extent of loss and time to olfactory recovery following infection is less well known. Similarly, knowledge of potential impactful patient factors and therapies that influence olfactory recovery is desirable but is not overtly clear in the literature. Our systematic review sought to fill this knowledge gap. We included studies that: involved either an observational or an interventional design that reported data on patients with olfactory dysfunction due to Reverse Transcription Polymerase Chain Reaction (RT-PCR) diagnosed COVID-19 infection; and reported data regarding olfactory recovery measured by an objective olfactory test, Likert scale and/or visual analog scale (VAS). The study methods were determined a priori and registered in PROSPERO (Registration Number CRD42020204354). An information specialist searched Medline, Embase, LitCovid and the Cochrane Register of Controlled Trials up to March 2021, and two reviewers were involved in all aspects of study selection and data collection. After screening 2788 citations, a total of 44 studies of assorted observational designs were included. Patients had undergone objective COVID-19 testing, and most were adult patients with mild to moderate COVID-19. Olfactory recovery was found to occur as early as 7 days, with most patients recovering olfaction within 30 days. Few studies included prolonged follow-up to 6 months or longer duration. Poor olfaction at initial presentation was associated with poor recovery rates. Only a small number of studies assessed olfactory retraining and steroid therapy. Additional trials are underway.

## Introduction

The spread of Coronavirus (COVID-19) infection was announced by the World Health Organization to be a pandemic on March 11th 2020 [1]. Since the pandemic began, more than 178

**Competing interests:** The authors have declared that no competing interests exist.

million confirmed cases of COVID-19 worldwide were reported as of June 22nd 2021 [2]. Amongst the most common symptoms of COVID-19 were fever, cough, shortness of breath, and myalgia [3, 4]. Anosmia (absence of olfaction) represents another common symptom, and may sometimes be the primary presenting symptom or the sole manifestation of disease in patients with COVID-19 [5, 6]. Several cross-sectional studies have been performed to assess otolaryngologic symptomatology prevalence rates amongst patients with COVID-19 infection [7]. A high prevalence rate of olfactory dysfunction and gustatory dysfunction was found to be reported in the literature [6]. Post-viral olfactory loss has been reported with Severe Acute Respiratory Syndrome (SARS), Middle East Respiratory Syndrome (MERS), influenza, parainfluenza, Respiratory Syncytial Virus (RSV), and adenovirus, however, the prevalence of olfactory loss is lower compared to COVID-19 [7]. The pathophysiology of olfactory loss following COVID-19 is not well understood, but a speculative mechanism is due to direct injury by the virus to olfactory receptor neurons located in the olfactory epithelium [8]. Smell and taste are essential sensory functions, and thus recovery of olfactory function has an important impact on patients' quality of life [9]. A previous qualitative review found that most studies on olfactory recovery were performed in cohorts of patients with olfactory loss following COVID-19 using subjective assessments in the forms of online questionnaires or telephone calls, however more recent studies have utilized objective testing tools of olfactory loss with measurement at uniform follow-up intervals [10]. To our knowledge, no recent review has sought to systematically compile this literature. To address this knowledge gap, we planned a systematic review to evaluate the extent and timing of olfactory recovery following loss due to COVID-19 infection.

## Review methods

The methods for the review were established a priori. This systematic review has been reported according to guidance from the Preferred Reporting Items for Systematic Reviews and Meta-Analyses (PRISMA) [11] The protocol was registered in PROSPERO on August 17th 2020, under Registration number CRD42020204354.

### Search strategy

Searches to identify relevant studies for this review were conducted by an experienced medical information specialist (RS) in consultation with the study authors. Literature searching was conducted in the following databases: Ovid MEDLINE, Embase, the Cochrane Central Register of Controlled Trials (CENTRAL) and LitCovid. The MEDLINE search strategies were peer reviewed by a senior information specialist (BS) using the PRESS checklist prior to extraction [12]. Systematic reviews on this topic were also retrieved from MEDLINE and their reference lists were scanned for additional potentially eligible studies to be screened. The searches were initially conducted on July 21st, 2020, were last updated in May 2021. A copy of the search strategies is provided in the Review Supplement (S1 Text).

### Study eligibility criteria

Study selection criteria were established according to considerations pertaining to population, study design and outcomes. The details of these criteria are described next.

**Population.** Studies were of interest if they enrolled individuals diagnosed with COVID-19 infection by positive PCR test, and who reported olfactory dysfunction (anosmia or hyposmia). Studies with data available for only a subgroup of patients with olfactory dysfunction were also included. All non-human studies were also excluded.

**Study design.** Studies involving either an observational or interventional design were of interest. No restrictions related to follow-up duration were employed. We excluded all review articles, case reports, letters, editorials, commentaries, and abstracts.

**Outcomes.** The primary outcome of interest was the extent of olfaction recovery following COVID-19 infection, using objective or subjective measures. Therefore, a baseline measurement of olfactory loss at time of infection or at study intake was compared to subsequent measurements at follow up intervals. Olfactory loss could be reported as a continuous or categorical measurement (anosmia, hyposmia, normosmia). Our secondary outcomes of interest included time to olfactory recovery, patient prognostic factors (for occurrence and resolution of olfactory dysfunction), and interventional therapies used for olfactory recovery.

Inclusion was limited to studies published in either the English, French, or Spanish languages.

## Process of study selection

The search yield from the literature search was imported into COVIDENCE software (Veritas Health Innovation, Version v2632 03f7ed40) [13]. Duplicate citations were removed prior to the start of screening. Selection of papers was conducted in two phases involving two independent reviewers (AJ and AL). During the first phase, each reviewer screened all titles/abstracts to make judgements as to relevance to the objectives and any conflict in the decision to include for further assessment was resolved by consensus discussion or consultation of a third party (SK). If the content of the abstract was unclear, the article was selected for full text review. In phase 2, full text articles of the citations from phase 1 deemed potentially eligible were screened using our predetermined inclusion and exclusion criteria. Any conflicts in the final decision to include a study were again resolved through consensus discussion or consultation of a third party (SK). The final process of study selection has been summarized using a PRISMA flow diagram (Fig 1).

## Process of data extraction

Analogous to our study selection process, two reviewers (AJ, AL) extracted data independently from the set of included studies using standardized data extraction form implemented in COVIDENCE software [13]. Discrepancies were settled by consensus discussion amongst the two reviewers. The data were then exported to Microsoft Excel (Version 16.49; Microsoft Corporation, Seattle, Washington, USA) for further preparation and to allow for inspection by team members to consider between study heterogeneity and study findings.

The following data were extracted from each study:

1. Identification: study ID (if available), citation, contact author details, country (or countries) of conduct of research origin

2. Methods: study design, total study duration, sample size, stated study objective, approach to data analysis, outcome definitions and measurement tools

3. Participant traits: mean (SD) patient age (or median/range if mean unavailable), race/ethnicity, diagnostic method for COVID-19, patient setting (non-hospitalized, hospitalized, medical care requiring intubation), severity of clinical manifestation of COVID-19 infection (as reported in the study), baseline measures of olfactory function

4. Outcomes: objective measures of olfactory dysfunction and recovery, timing for evaluation of olfactory dysfunction, scoring and interpretation of the instrument, therapies used for treatment of COVID-19 or the olfactory loss specifically, risk/protective factors associated

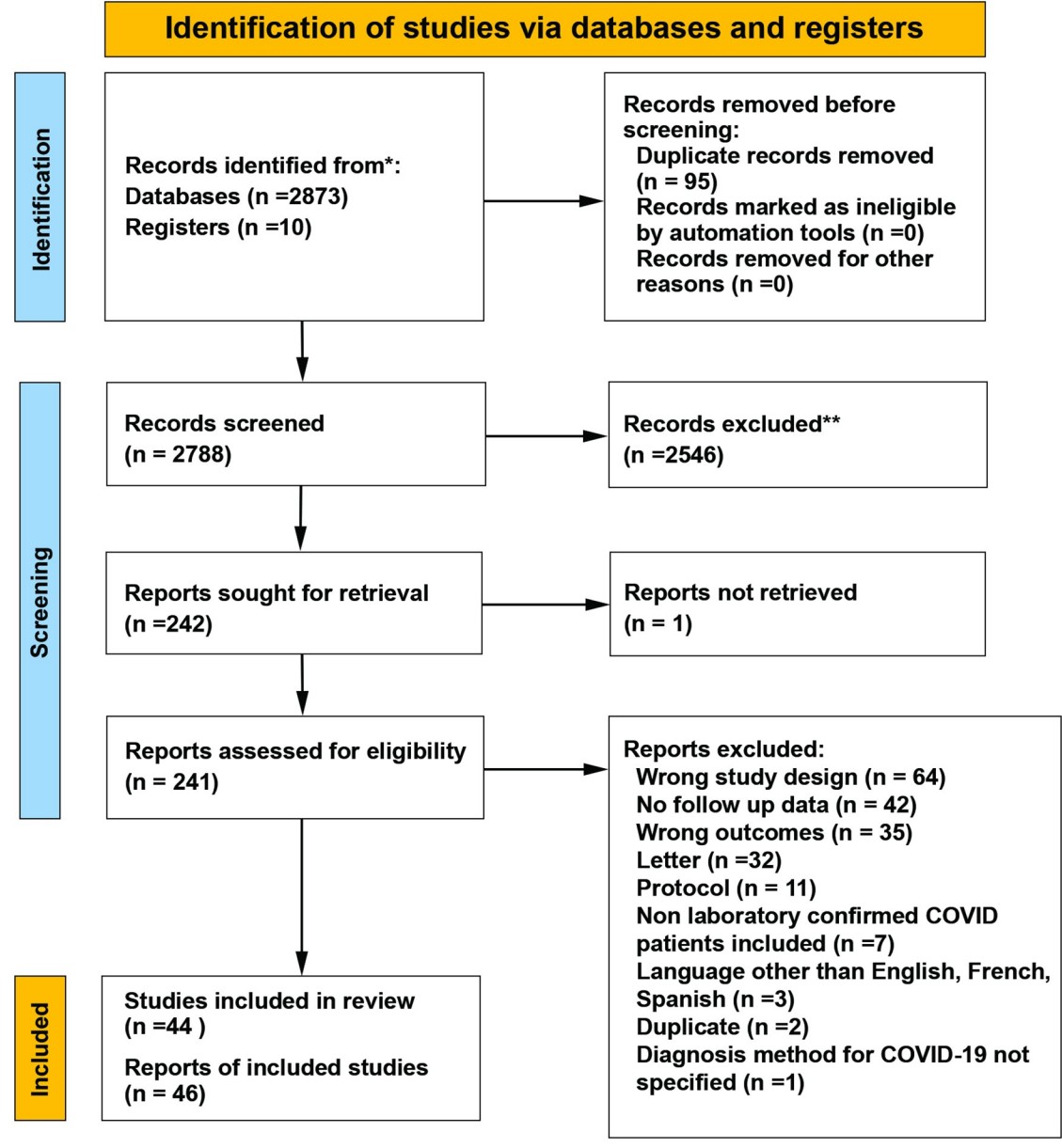

**Fig 1. PRISMA 2020 flow diagram describing the study selection process.**

with development of olfactory dysfunction associated with COVID-19, and/or its resolution.

5. Results: measures of olfaction dysfunction according to each instrument at available time points, length of time to recovery, listings of treatments administered to treat olfactory dysfunction, association measures for risk/protective factors for the development and/or resolution of olfactory dysfunction

6. Key conclusions of the study authors

7. Funding and declaration of conflict of interest.

### Risk of bias assessment

The risk of bias of the included studies was evaluated using relevant tools developed by the Joanna Briggs Institute (JBI) critical appraisal checklist [14]. The JBI assessment tool has separate assessments for each study design included in our systemic review. Case series as well as cross-sectional, cohort, and case-control studies were evaluated using the appropriate domains as set per the JBI assessment tools. Two reviewers (AJ, AL) independently evaluated the risk of bias of eligible studies and any disagreements were resolved by consensus discussion. Results from the risk of bias assessments were summarized descriptively to present key strengths and weaknesses of the included studies. The detailed findings from risk of bias assessments are presented in the supplemental files to this review.

### Approach to data synthesis

Due to considerable heterogeneity in study/patient characteristics and endpoints, no meta-analyses of outcomes were performed. A descriptive approach to summary of outcomes of interest was instead prepared. Grouping of studies has been based upon the type of olfactory assessment tool used, looking separately at studies that examined objective versus subjective measures. Study data have been organized and presented in table format which maps the outcomes reported by individual studies. The aggregate data of each study characteristic were reported, as well as testing instrument, initial olfaction scores, follow up time points, and prognostic factors as well as any therapeutics implemented. Combination of graphical and numerical displays of findings are presented for outcomes where appropriate.

## Results

### Study selection process

Our search identified a total of 2,788 citations for review following removal of duplicates. Amongst them, 2,546 were found to be irrelevant, and the remaining 241 were assessed for eligibility in full text. Overall, 46 publications describing 44 unique studies met our inclusion criteria, and 200 were excluded for reasons detailed in the flow diagram in Fig 1.

### Study characteristics

Details of the included studies are summarized in Table 1 (studies involving objective olfactory measures) and Table 2 (studies involving subjective olfactory measures), respectively. Table 3 presents the details of the studies that used objective and subjective olfactory measures. We identified variability amongst studies on type of measurement instrument adopted to assess olfactory function, the initial timing of assessment for olfactory dysfunction, the follow-up evaluation tools and intervals.

All studies were published since the start of the 2020 pandemic, and sample sizes ranged between 7 and 1363 participants. Most of the studies (n = 26, 59%) were conducted in Europe, including Italy (n = 11, 25%) [19, 20, 23, 28, 35, 36, 39, 47, 59], France (n = 3) [40, 44, 56], Germany (n = 3) [16, 17, 57], Belgium (n = 3) [25, 58, 60], Spain (n = 2) [43, 45], Greece (n = 1) (30) and Switzerland (n = 1) [31]; 3 European studies did not specify a country of origin [24, 46, 54]. The remaining 18 studies were conducted in other regions (a pie chart showing distribution is included in the supplementary data provided—S1 Fig). 10 studies were conducted in Asia: Turkey (n = 5) [18, 27, 38, 41, 50], Iran (n = 2) [22, 55], India (n = 2) [33, 52], Israel (n = 1) [32]. Four studies were conducted in North America: United States (n = 4), [21, 42, 48, 51]. Two studies were conducted in Africa: Egypt (n = 2) [34, 49], and two in South America: Brazil (n = 1) [53], Chile (n = 1) [15, 53]. Of the 44 studies, twenty-five (56.8%) reported no

**Table 1. Characteristics of studies that used objective olfactory measures.**

| First Author, Year | Country | Sample Size | Age (years) (mean/ median* (SD)and/or range) | COVID-19 Disease Severity | Olfactory Assessment Tool | Initial Smell Function | | |
|---|---|---|---|---|---|---|---|---|
| | | | | | | Anosmia | Hyposmia | Normosmia |
| | | | | | | % | % | % |
| **Cohort Studies** | | | | | | | | |
| Gonzalez, 2021 [15] | Chile | 100 | 42.1 (15.6) | Mild-Moderate | UPSIT | 12% | 63% | 25% |
| Niklassen, 2021 [16] | EU (Italy, Germany) | 111 | 44.5 (15) | Mild-Moderate | Sniffin' Sticks | 21% | 49% | 30% |
| Otte, 2021 [17] | EU(Germany) | 26 | 45 SE (2.06) | Mild-Moderate | Sniffin' Sticks | 0% | 100% | 0% |
| Ugurlu, 2021 [18] | Turkey | 42 | 41.2 (14.6) | Mild-Moderate | BSIT | 16.70% | 83.4% | 0% |
| Vaira, 2020 [19] | EU (Italy) | 345 | 48.5 (12.8) | Asymptomatic-Mild-Moderate | CCCRC test | NR | NR | NR |
| Vaira, 2020–3 [20] | EU (Italy) | 138 | 51.2 (8.8) | Mild-Moderate | CCCRC test | 40% | 27% | 33% |
| **Cross-sectional Studies** | | | | | | | | |
| Asad, 2021 [21] | United States | 14 | NR | Mild-Moderate | UPSIT | NR | NR | NR |
| **Case-Control Studies** | | | | | | | | |
| Moein, 2020 [22] | Iran | 100 | 45.40 (11.80) | Mild-Moderate | UPSIT (Persian version) | 18% | 78% | 4% |
| Vaira, 2020–2 [23] | EU (Italy) | 18 | 42.1 (NR) | Mild-Moderate | CCCRC test | NR | NR | NR |
| **Case Series** | | | | | | | | |
| Klimek, 2020 [24] | EU | 7 | NR (24–32) | Mild | Sniffin' Sticks | 14.30% | 85.7% | 0% |
| Le Bon, 2021 [25] | Europe (Belgium) | 27 | Steroid + OT group = 42 (14)<br><br>OT only group = 44 (14) | Mild-Moderate | Sniffin' Sticks | 22.22% | 77.77% | 0% |
| Petrocelli, 2021 [26] | Europe (Italy) | 300 | 43.6 (12.2) | Mild-Moderate | Self administered psychophysical evaluation (ethyl alcohol olfactory thresholds) | 47% | 16.30% | 36.70% |
| Salcan, 2021 [27] | Turkey | 94 | 53 (19.6) | Mild-Moderate | Sniffin' Sticks | NR | 71.2% | 28.8% |
| Vaira, 2020–1 [28] | EU (Italy) | 106 | 49.6 (8.5) | Mild-Moderate | CCCRC test | 41.50% | 25.4% | 33% |

SD = Standard Deviation; SE = Standard Error; NR = Not Reported; RT-PCR = Reverse Transcription–Polymerase Chain Reaction; UPSIT = The University of Pennsylvania Smell Identification Test; BSIT = The Brief Smell Identification Test; CCCRC = Connecticut Chemosensory Clinical Research Center Test; OT = Olfactory Training.

funding requirements for the project [16, 18, 19, 22, 27–30, 32, 33, 39, 40, 42–47, 49–51, 53, 58]; six (13.6%) studies reported funding for the project (sources included academic institutions, government and not-for-profit organizations) [15, 24, 25, 31, 48, 57], and thirteen (29.5%) did not report whether funding was provided [17, 21, 23, 26, 37, 38, 52, 54–56, 58, 59].

## Study population

All studies consisted of a patient population of asymptomatic or mild to moderate COVID-19 symptoms. Not all studies reported the scale used to classify the severity of COVID-19 but, in general, asymptomatic patients and those with mild symptoms did not require medical intervention or hospitalization (other than for isolation reasons). Patients with moderate symptoms required medical intervention but not admission to an Intensive Care Unit. The hospitalized patients included in all studies were of mild to moderate symptoms, none of which required invasive ventilation in ICU. COVID-19 was confirmed by RT-PCR test in all but six studies that either used RT-PCR or serology as a diagnostic tool of COVID-19 [25, 48, 51, 54, 58, 60]. Although most of the patients recruited within studies were not hospitalized due to COVID-19 infection [17, 21–26, 28, 30–32, 35, 36, 41–43, 45, 47–49, 51, 54, 56, 60], some did include patients hospitalized either for monitoring purposes or quarantine [18, 19, 33, 52, 55, 57–59]. Twelve studies included a mixture of hospitalized and non hospitalized patients [15, 19, 20, 27,

**Table 2. Characteristics of studies that used subjective olfactory measures.**

| First Author, Year | Country | Sample Size | Age (years) (mean/ median* (SD)and/or range) or [IQR] | COVID-19 Disease Severity | Olfactory Assessment Tool | Initial Smell Function | | |
|---|---|---|---|---|---|---|---|---|
| | | | | | | Anosmia | Hyposmia | Normosmia |
| | | | | | | % | % | % |
| **Cohort Studies** | | | | | | | | |
| Locatello, 2021 [29] | Europe (Italy) | 43 | 57.8 (NR) | Mild-Moderate | Chemosensory Complaint Score (CCS) | NR | NR | NR |
| Konstantinidis, 2020 [30] | EU (Greece) | 79 | 30.7 (5.3) | Mild-Moderate | VAS | NR | 36.70% | 63.30% |
| Speth, 2020 [31] | EU (Switzerland) | 112 | 46.8 (15.9) | Mild | Adapted Categorical Score System | 81.50% | 18.40% | 41% |
| **Cross-sectional Studies** | | | | | | | | |
| Biadsee, 2021 [32] | Israel | 97 | 37.5 (19–74) | Mild-Moderate | VAS | NR | 67% | 33% |
| Sahoo, 2021 [33] | India | 77 | 31.6 (9.6) | Mild-Moderate | Likeart scale (0–5) | 3% | 10.70% | 89.30% |
| Amer, 2020 [34] | Egypt | 96 | 34.26 (11.91) | Mild-Moderate | Adapted DyNaCHRON | NR | NR | NR |
| Boscolo-Rizzo, 2020 [35–37] | EU (Italy) | 183 | 56* [20–89] | Mild | SNOT-22 | 22.40% | 37.70% | 39.90% |
| Bulgurcu, 2020 [38] | Turkey | | 46.50 (15.20) | Mild-Moderate | VAS | 26.30% | 11.76% | NR |
| Dell'Era, 2020 [39] | EU (Italy) | 355 | 50* [49–59.5] | Asymptomatic or mild/moderate | Numerical Rating Scale | NR | NR | NR |
| Gorzkowski, 2020 [40] | EU (France) | 140 | 39.9 (13.7) | Mild-Moderate | Subjective Olfaction Numerical score | 64.29% | 35.71% | 0% |
| Sakalli, 2020 [41] | Turkey | 172 | 37.8 (12.5) | Mild-Moderate | Severity Scale (none, mild, moderate, severe) | 36% | 15.2% | 48.8% |
| Yan, 2020 [42] | USA | 23 | NR | Mild | VAS | NR | NR | NR |
| **Case-Control Studies** | | | | | | | | |
| Riestra-Ayora, 2021 [43] | Europe (Spain) | 320 (195 cases, 125 controls) | Cases = 41.62 (18–65) Control = 46.5 (20–64) | Mild-Moderate | VAS | NR | 64.1% | 35.9% |
| Eliezer, 2020 [44] | EU (France) | 40 | 34.6 (8.8) | Mild-Moderate | VOS | NR | NR | 0% |
| Martin-Sanz, 2020 [45] | EU (Spain) | 355 | 42.9 (0.67) | Mild-Moderate | VAS | 38.3% | 5.1% | NR |
| **Case Series** | | | | | | | | |
| Chary,2020 [46] | EU | 115 | 47* [20–83] | Mild | DyNACHRON | NR | NR | NR |
| Freni, 2020 [47] | EU (Italy) | 50 | 37.7 (17.9) | Mild-Moderate | sQOD-NS | NR | NR | NR |
| Janowitz, 2020 [48] | USA | 9 | NR (20–70) | Mild | 4-point adapted ECOG scale | 55.50% | 22.20% | 22.20% |
| **Miscellaneous Study Designs** | | | | | | | | |
| Abdelalim, 2021 [49] | Egypt | 100 | 29.0* [21.75–38.0] | Mild-Moderate | VAS | NR | NR | NR |
| Kavaz, 2021 [50] | Turkey | 53 | 42.7 (14.19) | Mild-Moderate | VAS | NR | 60.40% | 39.60% |
| Raad, 2021 [51] | USA | 343 | 43.5 (13) | Mild | SNOT–22 Rhinologic domain | NR | NR | NR |
| Yadav, 2021 [52] | India | 28 | 43.03 (16.10) | Mild-Moderate | sQOD-NS | 6.58% | 11.84% | 81.58% |
| BrandaoNeto, 2020 [53] | Brazil | 655 | 37.7 (10.4) | Mild-Moderate | VAS | 60.90% | 19.80% | NR |
| Chiesa-Estomba, 2020 [54] | EU | 751 | 41 SD +/- 13 | Mild | DyNaCHRON | 83% | 17% | 0% |
| Jalessi, 2020 [55] | Iran | 92 | 52.94 (13.25) | Mild-Moderate | Likeart scale (0–5) | 9.80% | 14.10% | 76.10% |

*(Continued)*

**Table 2.** (Continued)

| First Author, Year | Country | Sample Size | Age (years) (mean/median* (SD)and/or range) or [IQR] | COVID-19 Disease Severity | Olfactory Assessment Tool | Initial Smell Function | | |
|---|---|---|---|---|---|---|---|---|
| | | | | | | Anosmia | Hyposmia | Normosmia |
| | | | | | | % | % | % |
| Renaud, 2020 [56] | EU (France) | 97 | 35* [20–73] | Mild | 5-point severity scale | 60% | 40% | 0% |

SD = Standard Deviation; IQR = Interquartile Range; NR = Not Reported; RT-PCR = Reverse Transcription–Polymerase Chain Reaction; VAS = Visual Analogue Scale; VOS = Visual Olfactive Score; SNOT-22 = Sino-nasal Outcome Test-22; DyNaCHRON = Dysfonctionnement Nasal Chronique; sQOD-NS = short version of the Questionnaire of Olfactory Disorders-Negative Statements.

29, 34, 38–40, 46, 50, 53]. The majority of studies did not report data on ethnicity. In 29 studies the proportion of patients with anosmia, hyposmia and normosmia was available [15–18, 20, 22, 24–26, 28, 30–36, 40, 41, 43, 45, 48, 50, 52, 55–60]. In eight studies, all patients had olfactory dysfunction at study baseline [17, 18, 24, 25, 34, 40, 56, 57].

## Study designs

Amongst the 44 included studies, as shown in the Tables 1–3, we identified the following study designs: cohort (n = 12) [15–18, 20, 28–31, 57–59], cross-sectional (n = 11) [21, 32–36, 38–42], case-controls (n = 5) [22, 23, 43–45], case series (n = 4) [24, 46–48], and other miscellaneous study designs (n = 12) [49–52, 54–56]. Nine of the 44 studies were retrospective and evaluated patients at only one time point [19, 32, 38–40, 42, 48, 50, 56]. In studies that prospectively evaluated olfactory dysfunction, the timing of such evaluations varied greatly. The initial evaluations of olfaction were measured from the onset of either olfactory or general symptoms [15, 20, 25, 26, 28, 31, 43–45, 53, 60], from the time of positive COVID-19 test [16, 21, 23, 33, 34, 38, 45, 46, 57, 59], from time of infection [24, 51], during active phases of symptomatology [47], from the time of the swab [35, 36], at initial home/clinic visit [27, 30], at the time of admission to hospital [29, 52], during the recovery period [17, 22], after recovery/discharge

**Table 3. Study characteristics of studies that used objective and subjective olfactory measures.**

| First Author, Year | Country | Sample Size | Age (years) (mean/median* (SD) | COVID-19 Disease Severity | Olfactory Assessment Tool | Initial Smell Function | | |
|---|---|---|---|---|---|---|---|---|
| | | | | | | Anosmia | Hyposmia | Normosmia |
| | | | | | | % | % | % |
| **Cohort Studies** | | | | | | | | |
| Bertlich, 2021 [57] | Europe (Germany) | 23 | 59.0 (16.6) | Mild-Moderate | BSIT | 72.70% | 27.3% | 0% |
| | | | | | SNOT-22 | | | |
| | | | | | VAS | | | |
| Lechien, 2021 [58] | Europe | 233(Objective). 1363 (Subjective). | 44.5 (16.4) | Mild-Moderate | Sniffin' Sticks | 32.20% | 18.40% | 49.40% |
| | | | | | SNOT-22 | | | |
| Iannuzzi, 2020 [59] | EU (Italy) | 34 | 47.47 (13) | Mild-Moderate | Sniffin' Sticks, VAS | 10% | 53.30% | 36.70% |
| | | | | | HRS | | | |
| Lechien, 2020 [60] | EU (Belgium) | 88 | 42.6 (11.2) | Mild-Moderate | Sniffin' Sticks SNOT-22, | 40% | 35% | 25% |
| | | | | | sQOD-NS, NHANES (smell/taste components), tests | | | |

SD = Standard Deviation; RT-PCR = Reverse Transcription–Polymerase Chain Reaction; BSIT = The Brief Smell Identification Test; VAS = Visual Analogue Score; SNOT-22 = Sino-nasal Outcome Test-22; HRS = Hyposmia Rating Scale; sQOD-NS = short version of the Questionnaire of Olfactory Disorders-Negative Statements; NHANES = National Health and Nutrition Examination Survey.

[49], or at other specified times [41, 54, 55]. The timing and number of follow-up evaluations also varied amongst studies. The majority of studies performed one follow-up olfactory re-test following their first initial testing, and thirteen studies performed more than one follow-up evaluation [20, 23, 24, 26, 28, 34–36, 43, 51, 52, 57, 60].

## Study outcomes

**Objective outcome measures.** Objective methods were used to assess olfactory loss in 18 studies (40.9%) of studies (Table 1). The Sniffin' Sticks test was used in eight studies (n = 8) [16, 17, 24, 25, 27, 58–60], the Connecticut Chemosensory Clinical Research Center (CCCRC) test was used in four studies (n = 4) [19, 20, 23, 28], and the University of Pennsylvania Smell Identification Test (UPSIT) was performed in three studies (n = 3) [15, 21, 22]. Other objective testing included the Brief Smell Identification Test (BSIT) (n = 2) [18, 57], and one study measured the ethyl alcohol olfactory threshold by administering a self psychophysical test [26]. The timing of the assessments varied. Twelve studies performed their initial assessment ≤ 15 days after symptom onset or COVID-19 diagnosis [15, 16, 18, 20–22, 24, 26–28, 57, 60] while 4 studies performed the initial assessment 25 or more days after symptom onset or COVID-19 diagnosis [17, 23, 25, 59]. Timing of follow-up evaluations ranged from 10 days [27] to 6 months [17, 57, 60].

**Subjective outcome measures.** Subjective methods of smell evaluation using objective scoring methods were used in 68% (n = 30) studies, as can be seen in Table 2. The most common subjective method to evaluate smell dysfunction was use of a Visual Analogue Scale (VAS) [19, 30, 32, 38–40, 42–44, 49, 50, 53, 57, 59]. Other methods included Likert scales [31, 33, 41, 48, 55, 56, 59], the Questionnaire of olfactory disorders–negative statements (sQOD-NS) [40, 47, 52, 54, 55, 58, 60], the DyNACHRON questionnaire (only the smell related questions) [34, 46, 54], and the Sinonasal outcome test– 22 (SNOT-22) [35, 36, 51, 57, 58, 60]. Table 3 shows the studies that used both objective and subjective methods to evaluate smell dysfunction [19, 20, 28, 29, 44].

## Findings from risk of bias appraisals

The evaluations using the JBI critical appraisal checklists showed several limitations in the available evidence. Risk of bias was considered high for the studies that performed the evaluation of olfactory function at a single point in time given the risk for recall bias [19, 38–40, 42, 48, 50, 56]. The only randomized controlled trial included in this review did not blind participants or clinicians to the assignment of participants to the use of therapy, nasal steroids, therefore the risk of bias for this study was considered high [49]. In one pilot study, participants were not randomly assigned to the study arm, which has the potential to introduce selection bias [25]. S1 Table shows the results of the evaluations for each study.

## Extent and time to recovery

Reporting of initial baseline anosmia and hyposmia was inconsistent amongst included studies. In those studies that performed objective testing, less than 40% of the population reported normosmia at initial evaluation compared to those who performed subjective testing, where normosmia could range from 0% to 90% at initial evaluation. Studies with follow up data regarding the extent of recovery of olfactory function and the time to recovery were analysed. A total of 4 studies that used objective olfactory measures completed follow-up data at 1 month [15, 24, 27, 28]; rates of full olfactory recovery in these studies ranged from 44.3% to 94.6% (median 72.6%). Seven studies completed follow-up data for up to 60 days [16, 20–23, 59, 60], and rates of full recovery in these studies ranged from 0% to 79.5% (median 73.3%).

Six studies completed follow-up data for more than 90 days and up to 6 months [17, 18, 25, 26, 57, 58]. The rates of full recovery in these studies ranged from 58.8% to 85.7% (median 73.5%). A total of 10 studies that performed subjective olfactory measures completed follow-up data at 1 month [30, 33, 34, 41, 44–47, 52, 55] with rates of full recovery in these studies ranging from 22.7% to 100% (median 73%); six studies completed follow-up between 30 and 60 days [29, 31, 35–37, 46, 49, 53], the recovery rates in these studies ranged from 53.8% to 81.4% (median 62.3%). Two studies completed follow-up at 6 months [43, 51] only one reported rate of complete recovery at 58.4% (43). The study-specific findings for these measures are provided in Tables 4 and 5.

The frequently observed mean days to olfactory recovery was within 2 weeks [18, 35, 41, 42, 46, 54, 56, 58], with some studies reporting recovery continuing up to 30 days [15, 20, 23, 24, 27, 29, 34, 51] and those with longer duration of follow-up to 6 months reporting most recovery to occur within 60 days of olfactory dysfunction, yet in other studies recovery was observed to continue up to 6 months [26, 43].

## Protective and risk factors for olfactory dysfunction and resolution

Approaches to explore associations between olfactory resolution and other patient characteristics varied between studies. This included both univariate and multivariable regression analyses, subgroup analyses, and correlation studies. Amongst the most commonly studied factors were age, gender, medical co-morbidities, and initial severity of smell loss. Studies evaluating the effects of age determined it was not a factor that contributed to poor recovery rate [16, 18–20, 25, 29, 46, 60]. One study showed age was inversely associated with improvement of olfactory function [15]. A positive association between age and the duration of hyposmia/anosmia was reported by one study [49] another study reported that persistent olfactory dysfunction is associated with age ($\geq$ 50 years) [26]. When the effect of gender on olfactory recovery was assessed, the majority of studies did not show any significant relationship between recovery and gender [15, 16, 26, 29, 32, 38, 46, 60], while one study found that females represent more than 70% of those who completely recovered [34] and one study showed female gender showed slower olfactory recovery rates [53]. Many studies had specific population eligibility criteria, thus excluding patients with an a priori diagnosis of neurodegenerative disease, asthma, craniofacial trauma, CRS, nasal polyposis disease. However, two studies identified that diabetes mellitus diagnosis amongst patients was a poor prognostic factor for recovery [34, 49] while others did not identify medical co-morbidities including diabetes mellitus to be have any prognostic value [19, 20, 31, 54]. An observational negative prognostic factor commonly identified was the initial degree of smell loss. Patients with more severe olfactory dysfunction at initial presentation had worse olfactory recovery outcomes compared to those with very mild to moderate hyposmia at initial presentation [18, 20, 30, 54]. S2 Table presents a summary of studies that assessed the effects of different characteristics, as well as the nature of the association that was reported by the study authors.

## Treatment intervention(s)

There was a paucity of studies in the literature that assessed interventions for the treatment of olfactory dysfunction following COVID-19 infection. A randomized controlled trial by Abdelalim et al 2021 [49] compared topical corticosteroid nasal spray (2 puffs, 100 mcg, mometasone furoate each nostril) once daily for 3 weeks in addition to olfactory training vs olfactory training only. This study included 100 patients who had recovered from COVID-19, demonstrated by 2 negative RT-PCR tests, but who had not recovered their olfactory function; Olfactory dysfunction (OD) was measured using a VAS. The study concluded that the use of nasal

**Table 4. Olfactory recovery at the end of follow up in studies that used objective olfactory measures.**

| Study ID | Sample Size | Objective measures of olfactory dysfunction and recovery | Timing of evaluations | End of F/U Anosmia (%) | End of F/U Hyposmia (%) | End of F/U Normosmia (%) |
|---|---|---|---|---|---|---|
| **Follow up ≤30 days** | | | | | | |
| Klimek 2020 [24] | **7** | **Sniffin' Sticks** | * Initial: 1–14 days (during quarantine) | 14.30% | 0.00% | 85.7% |
| | | | *Test 2: Day 21 | | | |
| | | | *Test 3: Day 28 | | | |
| Vaira 2020–1 [28] | **106 patients (71 with olfactory dysfunction)** | **Self administered olfactory and gustatory psychophysical test CCCRC** | * Initial: Within 4 days of clinical onset | 8.40% | 47.1% (Severe hyposmia 11.3% + Moderate hyposmia 8.4% + Mild hyposmia 24.7%) | 44.3% |
| | | | * Test 2: 10 days | | | |
| | | | * Test 3: 20 days | | | |
| Salcan 2021 [27] | **94** | **Sniffin' Sticks** | *Initial: At first clinical encounter | - | 3 patients (5.4% of those with initial smell alteration with a second evaluation n = 55) | 94.6% |
| | | | *Test 2: 10 days from initial | | | |
| Gonzalez 2021 [15] | **100** | **UPSIT** | *Initial: < 15 days of ongoing symptoms | 0% | 40.5% (mild microsmia 22.1 + moderate microsmia 15.7% + severe microsmia 3.1%) | 59.5% |
| | | | *Test 1: 30 days after symptoms onset | | | |
| **Follow up 31–60 days** | | | | | | |
| Vaira 2020–3 [20] | **138** | **Self administered olfactory and gustatory psychophysical test CCCRC** | * Initial: Within 4 days of symptom onset | 1% | 20% | 79% |
| | | | * Test 2: 10 days after symptom onset | | | |
| | | | *Test 3–5: Every 10 days | | | |
| Iannuzzi 2020 [59] | **34** | **Sniffin Sticks** | *Initial: 25 days after COVID-19 diagnosis | 0 | 26.70% | 73.3% |
| | | | * Test 2: 1 month after initial | | | |
| Moein 2020 [22] | **100 patients (82 retested)** | **UPSIT** | *Initial: During recovery period. | 0% | 39% (Mild microsmia 20% + moderate microsmia 13% + severe microsmia 6%) | 61% |
| | | | *Test 2: 1 or 4 weeks | | | |
| Lechien 2020 [60] | **88** | **Sniffin' Sticks** | * Initial: When acute course of disease was resolved. | NR | NR | 79.5% |
| | | | * Test 2: 3–15 days from onset | | | |
| | | | *Test 3: 2 months | | | |
| Vaira 2020–2 [23] | **18** | **CCCRC** | * Initial: 30 days after diagnosis | Treatment Group: 0% | Treatment Group: 44.4% | Treatment Group: 55.5% |
| | | | * Test 2: 20 days | No treatment Group: 22.2% | No treatment Group: 77% | No treatment Group: 0% |
| | | | * Test 3: 40 days | | | |
| Niklassen 2021 [16] | **111** | **Sniffin' Sticks** | *Initial: Within 3 days of positive COVID | 1% | 25% | 74% |
| | | | * Test 2: Varied by study site. Average 62.9 days +/- 45.8 | | | |
| Asad 2021 [21] | **14 Cov 19 Positive patients** | **UPSIT** | *Initial: Within 48 hours of COVID-19 test | NR | NR | NR |
| | | | *Test 2: 6 weeks | | | |
| **Follow up ≥ 90 days** | | | | | | |

*(Continued)*

**Table 4.** (Continued)

| Study ID | Sample Size | Objective measures of olfactory dysfunction and recovery | Timing of evaluations | End of F/U Anosmia (%) | End of F/U Hyposmia (%) | End of F/U Normosmia (%) |
|---|---|---|---|---|---|---|
| Ugurlu 2021 [18] | 104 (42 with smell loss) | **BSIT** | *Initial: At diagnosis | - | 14.3% | 85.7% |
| | | | *Test 2: 3 months | | | |
| Petrocelli 2021 [26] | **300** | **Self-administered psychophysical evaluation** | *Initial: Within 7 days of OD | 5% | 22% | 73% |
| | | | *Test 2–4: 1,2,3 months *Test 5: 6 months | | | |
| Otte 2021 [17] | **26** | **Sniffin' Sticks** | *Initial: 3 months post COVID | 0% | 26.90% | 73.1% |
| | | | *Test 2: 6 months after infection | | | |
| LeBon 2021 [25] | **27** | **Sniffin' Sticks** | *Initial: Average 5 weeks after onset of OD | NR | NR | NR |
| | | | *Test 2: 10 weeks | | | |
| Bertlich 2021 [57] | **23** | **BSIT** | *Initial: After positive test. | 5.90% | 35.3% | 58.8% |
| | | | *Test 2: 8 weeks | | | |
| | | | *Test 3: 6 months | | | |
| Lechien 2021 [58] | **233** | **Sniffin' Sticks** | *Initial: 2–3 weeks from symptom onset | NR | 15.3% (not fully recovered) | NR |
| | | | *Test 2: 60 days | | | |
| | | | *Test 3: 6 months | | | |

UPSIT = The University of Pennsylvania Smell Identification Test; BSIT = The Brief Smell Identification Test; CCCRC = Connecticut Chemosensory Clinical Research Center Test; NR = Not Reported.

steroid spray showed no added benefit over olfactory training alone. A pilot study by LeBon et al 2021 [25] compared two groups; Group 1 received 32 mg of methylprednisone orally once daily combined with olfactory training twice daily for 10 weeks compared to olfactory training only. This study included 27 patients who had recovered from COVID-19 but showed persistent OD five weeks after the onset of symptoms. The Sniffin' Sticks test was used to measure OD. The outcome showed that combination therapy may be beneficial, although robust conclusions could not be made due to low recruitment of participants [25]. Vaira et al 2021 [23], conducted a case-control study in which patients were randomly assigned to systemic corticosteroids (prednisone starting with 1mg/kg day) tapering for 15 days as well as nasal irrigation with betamethasone, ambroxol and rinazine for 15 days or no treatment. This study included 18 patients who had recovered from COVID-19, demonstrated by 2 negative tests, and who still demonstrated anosmia or hyposmia 30 days after clinical onset measured using the CCCRC test. In this study, patients in the treatment group showed significantly higher improvement in the CCRC test at both the 20 and 40-day evaluations. We searched the ClinicalTrials.gov registry in March 2021 to identify ongoing clinical trials assessing the efficacy of treatment of olfactory dysfunction related to COVID-19. We identified 9 trials, (NCT04764981, NCT04710394, NCT04361474, NCT04657809, NCT04406584, NCT04495816, NCT04569825, NCT04789499, NCT04528329) and we have provided their details in S3 Table. These studies are evaluating the effects of therapies that include olfactory

**Table 5. Olfactory recovery at the end of follow up in studies that used subjective olfactory measures.**

| Study ID | Sample Size | Objective measures of olfactory dysfunction and recovery | Time of evaluations | End of F/U Anosmia (%) | End of F/U Hyposmia (%) | End of F/U Normosmia (%) |
|---|---|---|---|---|---|---|
| | | | **Follow up ≤30 days** | | | |
| Jalessi 2020 [55] | 92 | **Likert scale (0–5)** | *Initial: While in hospital | 0% | 4.55% (partial improvement) | 95.45% (complete resolution) |
| | | | * Test 2: Mean follow up 20.10 ± 7.42 days | | | |
| Freni 2020 [47] | 50 | **sQOD-NS** | * Initial: During active phase of symptoms | NR | NR | 82% (no olfactory dysfunction at follow up) |
| | | | * Test 2: 15 days after COVID test negative | | | |
| Chary 2020 [46] | 115 | **DyNaCHRON** | *Initial: Day 5 after COVID + test | NR | 33% (partial recovery) | 64% (complete recovery) |
| | | | * Test 2: Day 15 after COVID test | 3% (no recovery) | | |
| Sakalli 2020 [41] | 172 (88) had OD | **Scale for severity (none, mild, moderate, severe)** | * Initial: NR | NR | 21.6 (no recovery) 18.2% (mild recovery) 37.5% (moderate recovery) | 22.7% (complete recovery) |
| | | | * Test 2: 20 days after diagnosis | | | |
| Konstantinidis 2020 [30] | 79 | **VAS** | * Initial: Before test results reported | NR | 36.6% (partial or no improvement) of those with OD at baseline | 63.3% (almost complete recovery) |
| | | | *Test 2: 4 weeks | | | |
| Amer 2020 [34] | 96 | **DyNaCHRON** | * Initial: After positive test | NR 25% (no recovery) | 41.7% (partial recovery) | 33.3% (full recovery) |
| | | | * Test 2–5: weekly for 4 weeks | | | |
| Martin-Sanz 2020 [45] | 355 | **VAS** | *Initial: 7 days from symptom onset | NR | NR | 85.4% (recovered olfactory function) |
| | | | * Test2: 2 weeks | | | |
| Eliezer 2020 [44] | 40 | **Visual Olfactive Score (VOS) and Evaluation to assess the ability to detect and identify odorants** | * Initial: Within 15 days of OD onset | 0% | 60% (Olfactory score 1–9 at f/u) | 40% (Olfactory score = 10 at f/u) |
| | | | * Test 2: 1 month | | | |
| Yadav 2021 [52] | 152 (28 reported olfactory dysfunction) | **sQOD-NS** | *Initial: At admission | 0% | 0% | 100% (of the 28 with olfactory dysfunction at baseline) |
| | | | *Test 2: 7 days | | | |
| | | | *Test 3: 14 days | | | |
| | | | *Test 3: At discharge | | | |
| Sahoo 2021 [33] | 718 (77 reported smell loss) | **Likert scale** | *Initial: 3 days after + test | 0% (score = 5) | 7.8% (scores 1–4) | 92.2% (scores = 0) |
| | | | * Test 2: 2 weeks | | | |
| | | | **Follow up 30–60 days** | | | |
| Chary 2020 [46] | 115 | **DyNaCHRON** | *Initial: Day 5 + test | NR 3% (no recovery) | 33% (partial recovery) | 64% (complete recovery) |
| | | | * Test 2: Day 15 after + test | | | |
| Boscolo-Rizzo 2020 [35–37] | 183 | **SNOT-22** | *Initial: 5–6 days after swab | 0% (score = 5) | 18.6% (SNOT score = 1–4) | 81.4% (SNOT score = 0) After 8 weeks |
| | | | *Test 2: 4 weeks | | | |
| | | | *Test 3: 8 weeks | | | |

*(Continued)*

**Table 5.** (Continued)

| Study ID | Sample Size | Objective measures of olfactory dysfunction and recovery | Time of evaluations | End of F/U Anosmia (%) | End of F/U Hyposmia (%) | End of F/U Normosmia (%) |
|---|---|---|---|---|---|---|
| Speth 2020 [31] | Total 112 (66 had OD) | **At its worst scale of 0 (none), 1 (mild), 2 (moderate), or 3 (severe) Recovery: Full resolution partial resolution, No resolution** | * Initial: NR<br>*Test 2: 36 days from beginning of symptoms | NR | NR | 78.8% (who had COVID symptoms 22–60 days ago experienced complete resolution at a mean time of 11 days) |
| Locatello 2021 [29] | 43 | **CCS** | *Initial: On admission<br>* Test 2: 30 days after negative Test | NR | 39.4% (Persistent hyposmia) | 60.6% (Complete recovery) |
| Abdelalim 2021 [49] | 108 | **VAS** | *Initial: After recovery/ discharge<br>*Test 2: week 1<br>*Test 3: week 2<br>*Test 4: week 3 | NR | NR | Group 1: 62% of patients had sense of smell completely recovered. Group 2: 52% of patients had sense of smell completely recovered. All patients: 57% |
| Branda Neto 2020 [53] | 655 (545 recruited from social network, 110 were at hospital) | **VAS** | * Initial: 15 days after symptom onset<br>* Test 2: 36–119 days from symptom onset | NR (no recovery 1.4%) | 44.7% (partial recovery) | 53.8% (total recovery) |
| | | | **Follow up ≥90days** | | | |
| Riestra-Ayora 2021 [43] | 320 (195 in case group) | **VAS** | *Initial: At onset of symptoms<br>*Test 2: 1 month<br>*Test 3: 3 months *Test 4: 6 months | 11.2% (showed no recovery) | 30.4% (Partial recovery) | 58.4% (Complete recovery) |
| Raad 2021 [51] | 521 completed initial survey<br><br>343 completed 6 months follow up | **SNOT -22 rhinologic domain** | *Initial: 2 weeks prior to diagnosis, 2 weeks and 4 weeks after diagnosis (retrospective)<br>*Test 2: 6 months | NR | NR | NR |

F/U = Follow up; sQOD-NS = short version of the Questionnaire of Olfactory Disorders-Negative Statements; DyNaCHRON = Dysfonctionnement Nasal Chronique; VAS = Visual Analogue Scale; SNOT-22 = Sinonasal Outcome Test– 22; CCS = Chemosensory Complaint Score; NR = Not Reported.

training, systemic and intranasal corticosteroids, insulin fast dissolving film, theophylline nasal irrigations and injections of platelet rich plasma into the olfactory cleft.

## Discussion

In this study, the evidence regarding olfactory recovery following COVID-19 infection was reviewed. Our literature review identified 44 studies that met our inclusion criteria that used objective and/or subjective olfactory evaluation. We identified considerable heterogeneity between the studies in terms of study designs employed, psychophysical testing tools utilised, the timing of initial assessments, follow-up intervals, and recording of olfactory scores. Due to this high degree of between-study variability, we did not perform meta-analyses of outcome data, and results were presented using a descriptive approach.

Olfactory recovery most often occurred within the first two weeks from symptom onset with a rate of recovery at 1 month as high as 94.6%; at 6 months the rate of recovery was as high as 85.7%. We analyzed any treatment interventions as our secondary outcome. Only

three published papers assessed treatment for persistent OD due to COVID-19 [23, 25, 49]. The treatments included oral and nasal corticosteroids and olfactory training. The small numbers of studies evaluating treatments and the similarly small numbers of patients included in these studies limit the ability to make firm conclusions. However, it has been demonstrated that olfactory training is associated with clinical improvement of olfactory dysfunction following upper respiratory tract viral infection [61, 62] and therefore it is a reasonable treatment alternative. There are several ongoing studies (NCT04764981, NCT04710394, NCT04361474, NCT04657809, NCT04406584, NCT04495816, NCT04569825, NCT04789499, NCT04528329) evaluating olfactory training and once completed, they will provide insight on whether this treatment is also effective in olfactory dysfunction due to COVID-19. The use of corticosteroids in post viral olfactory dysfunctions is not widely recommended [62–64] and their use in OD due to COVID-19 needs further study. We identified that patients with initial poor olfactory scores had slow rates of recovery, however the majority of studies did not observe medical co-morbidities, demographics, or general symptoms to be a pertinent factor in olfactory recovery.

## Strengths, limitations and implications for clinical practice

More than a year has passed since the WHO declared the COVID-19 pandemic. Many countries have experienced multiple waves and more recently the surfacing of numerous widespread virus variants. Concerns over olfactory dysfunction have become a common complaint of many patients surviving COVID-19 and a source of referral to clinicians. The data presented in this review can be used by healthcare professionals to counsel patients regarding the duration and expected recovery from OD. The major strength of our study lies in the fact that all the studies included in this review used RT-PCR to confirm COVID-19. We included studies that used objective and/or subjective measures for olfactory dysfunction. It is well known that olfactory dysfunction may be underreported when using subjective methods and thus the use of objective tests is preferred. However, the information provided by the studies using subjective methods may be used by clinicians in settings where access to objective methods is limited.

There was considerably a high level of heterogeneity across studies that restricted our ability to carry out a pooled statistical analysis. Some studies, especially those from early in the pandemic, performed a retrospective assessment of olfaction which is subject to recall bias. Although we updated our literature search twice, the speed at which new studies related to COVID-19 are being published makes it difficult to include all available evidence in this review. Updates to this review will be needed in the future. We expect that newer studies will include randomized controlled trials evaluating therapies for OD with larger numbers of participants and longitudinal studies that will have longer follow up times.

## Conclusion

Our study identified several important and clinically relevant factors related to olfactory dysfunction following COVID-19 infection. We noted that the majority of studies showed olfactory recovery as early as the first two weeks, although some studies with more prolonged follow-up showed a small population of patients can have residual olfactory loss even after 6 months. Poor olfaction at initial presentation was associated with poor recovery rates, and factors such as age, gender, or medical co-morbidities did not play a major role in olfactory recovery. Treatment interventions included steroid therapy (topical/systemic) and/or olfactory retraining, but no robust conclusions were possible. Future research in this area should include large longitudinal studies that follow the course of the olfactory dysfunction beyond 6 months

and well designed RCTs that are powered to determine effectiveness of olfactory training and other therapies for the treatment of persistent olfactory dysfunction due to COVID-19.

## Supporting information

**S1 Fig. Publications by region of origin.**
(TIF)

**S1 Table. JBI critical appraisal checklist.**
(DOCX)

**S2 Table. Risk and protective factors.**
(DOCX)

**S3 Table. Ongoing studies evaluating treatments for olfactory dysfunction due to COVID-19 infection.**
(DOCX)

**S1 Checklist. PRISMA 2020 checklist.**
(DOCX)

**S1 Text. Literature search strategy.**
(DOCX)

**S2 Text. Studies excluded in the full text review.**
(DOCX)

**S3 Text. Protocol.**
(DOCX)

## Acknowledgments

We would like to acknowledge Ms. Becky Skidmore for her contribution to the literature search design and review.

## Author Contributions

**Conceptualization:** Ali Jafar, Brian Hutton, Shaun Kilty.

**Data curation:** Ali Jafar, Andrea Lasso.

**Investigation:** Ali Jafar, Andrea Lasso, Risa Shorr.

**Methodology:** Brian Hutton.

**Project administration:** Andrea Lasso.

**Resources:** Risa Shorr.

**Supervision:** Shaun Kilty.

**Writing – original draft:** Ali Jafar.

**Writing – review & editing:** Andrea Lasso, Brian Hutton, Shaun Kilty.

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
