## [Decision Letter · Decision Letter 0]

14 Sep 2021

PONE-D-21-25736Olfactory recovery following infection with COVID-19: A systematic reviewPLOS ONE

Dear Dr. Kilty,

Thank you for submitting your manuscript to PLOS ONE. After careful consideration, we feel that it has merit but does not fully meet PLOS ONE’s publication criteria as it currently stands. Therefore, we invite you to submit a revised version of the manuscript that addresses the points raised during the review process.

We look forward to receiving your revised manuscript.

Kind regards,

Eng Ooi

Academic Editor

PLOS ONE

Journal Requirements:

Additional Editor Comments:

The paper has been reviewed by 3 reviewers. They have mostly positive comments but have recommended minor revisions. Please address all their comments and suggested revisions.

Reviewers' comments:

Reviewer's Responses to Questions

**Comments to the Author**

1. Is the manuscript technically sound, and do the data support the conclusions?

Reviewer #1: Yes

Reviewer #2: Yes

Reviewer #3: Yes

2. Has the statistical analysis been performed appropriately and rigorously? 

Reviewer #1: N/A

Reviewer #2: Yes

Reviewer #3: Yes

3. Have the authors made all data underlying the findings in their manuscript fully available?

Reviewer #1: Yes

Reviewer #2: Yes

Reviewer #3: Yes

4. Is the manuscript presented in an intelligible fashion and written in standard English?

Reviewer #1: Yes

Reviewer #2: Yes

Reviewer #3: Yes

5. Review Comments to the Author

Reviewer #1: This is an in-depth systematic review looking at the recovery of olfactory dysfunction in patients with COVID – 19. I only have a few comments:

1. The number of records excluded/irrelevant, don’t match the text (page 8, line 194) and figure 1. In the text it is 2533, in figure 1 it is 2546. Is this an error?

2. Page 18, line 255 – what is the range of sample size? There is only 1 number: 1363, please correct

3. The authors mentioned the rates of full recovery of olfaction. Was there any data that could be extracted about the rates of partial recovery or some degree of recovery?

4. Were papers including children specifically excluded from the review?

Again a great paper and I commend the authors on the very thorough review.

Reviewer #2: Thank you for the opportunity to review this systematic review on an important topic.

Overall, this is a well conducted systematic review with a priori methods published.

You have clearly reported the search and study selection with independent reviewers to reduce bias.

Good use of JBI tools for study quality analysis. Descriptive analysis is appropriate as significant heterogeneity in study design and results.

There is a lot of good data presented in all the tables, however, they are all very busy. To allow easier perusal for reader, please reduce unnecessary words and use acronyms where appropriate.

Table 1 and Table 2 – “Not reported” in table 1, “NR” in table 2. Maintain consistent acronyms.

Please use one or the other

Table 3. Lechien 2021 (58) – Sample size. Reduce wording. Ie. 233 (Objective) 1362 (Subjective)

Study population

• Can you clarify the definition of mild vs mild-moderate symptom severity?

Line 298. The studies identified to have one test evaluating OD at a single point in time were deemed to have high level of bias. Please clarify in this section if these studies were excluded from analysis. As it would not be possible obtain follow up data with only one data point.

Table 4. The “Time of evaluation” column is very busy. Can this be simplified? For example.

• Test 1: Day 1.

• Test 2: Day 10.

• Test 3: Day 31

Table 5. If you could label “Time of evaluation” column the same as Table 4. Similar to table 4, simplify and reduce the wording in this column. This will make the follow up times easier to follow for the reader.

Reviewer #3: This paper adequately summarises the growing number of investigations into COVID-19 related olfactory dysfunction, and provides succinct evidence to counsel patients on the expected recovery of olfactory dysfunction. The article is technically sound, and the data supports the conclusions made.

The methods were appropriate, and the PRISMA reporting guideline for systematic reviews was followed.

All supplementary data was made available for review

There are no concerns regarding the methods, reporting of results, or discussion.

6. PLOS authors have the option to publish the peer review history of their article (what does this mean?). If published, this will include your full peer review and any attached files.

Reviewer #1: No

Reviewer #2: No

Reviewer #3: **Yes: **Jae Murphy

---

## [Author Response · Author response to Decision Letter 0]

1 Oct 2021

Response: Thank you for the comment. We have formatted the manuscript document to follow PLOS ONE’s style requirements. 

Response: Thank you for the comment. We have checked the references included in this paper and to the best of our knowledge none have been retracted. 

Response: Thank you for the comment. All the data required to replicate the results of this study is available as part of the supporting information.

REVIEWERS COMMENTS:

Reviewer #1: This is an in-depth systematic review looking at the recovery of olfactory dysfunction in patients with COVID – 19. I only have a few comments:

1. The number of records excluded/irrelevant, don’t match the text (page 8, line 194) and figure 1. In the text it is 2533, in figure 1 it is 2546. Is this an error?

Response: Thank you for your comment. The correct number is 2546, we have corrected this error. 

2. Page 18, line 255 – what is the range of sample size? There is only 1 number: 1363, please correct

Response: Thank you for your comment. This error has been corrected. The correct range is 7-1363 patients

3. The authors mentioned the rates of full recovery of olfaction. Was there any data that could be extracted about the rates of partial recovery or some degree of recovery?

Response: Thank you for your comment. Tables 4 and 5 show the proportion of patients who reported anosmia or hyposmia at the end of the follow up period. If studies reported data on partial improvement of no improvement/recovery this is specified for each study on these columns. 

4. Were papers including children specifically excluded from the review?

Response: Thank you for your question. The inclusion criteria for this review did not specifically exclude studies that included children. However, none of the included studies was conducted in pediatric population. 

Again a great paper and I commend the authors on the very thorough review.

Reviewer #2: Thank you for the opportunity to review this systematic review on an important topic.

Overall, this is a well conducted systematic review with a priori methods published.

You have clearly reported the search and study selection with independent reviewers to reduce bias.

Good use of JBI tools for study quality analysis. Descriptive analysis is appropriate as significant heterogeneity in study design and results.

1. There is a lot of good data presented in all the tables, however, they are all very busy. To allow easier perusal for reader, please reduce unnecessary words and use acronyms where appropriate.

Response: Thank you for your comments. We have edited the tables for better visualization and ensured that acronyms were used consistently. 

2. Table 1 and Table 2 – “Not reported” in table 1, “NR” in table 2. Maintain consistent acronyms.

Please use one or the other

Response: Thank you for your comment. We have updated to tables and “NR” has been used consistently.

3. Table 3. Lechien 2021 (58) – Sample size. Reduce wording. Ie. 233 (Objective) 1362 (Subjective)

Response: Thank you for your comment. We have updated Table 3 to include your suggestion. 

4. Study population

• Can you clarify the definition of mild vs mild-moderate symptom severity?

Response: Thank you for your comment. The study population section has been updated to clarify the definition of mild and moderate disease severity. 

5. Line 298. The studies identified to have one test evaluating OD at a single point in time were deemed to have high level of bias. Please clarify in this section if these studies were excluded from analysis. As it would not be possible obtain follow up data with only one data point.

Response: Thank you for your comment. These studies were not included. The section Extent and time to recovery has been updated to clarify this point. 

6. Table 4. The “Time of evaluation” column is very busy. Can this be simplified? For example.

• Test 1: Day 1.

• Test 2: Day 10.

• Test 3: Day 31

Response: Thank you for your comment. We have updated Table 4 and 5. We have also eliminated the Timing of evaluation in Tables 1-3 as this was the same information contained in Tables 4-5.

7. Table 5. If you could label “Time of evaluation” column the same as Table 4. Similar to table 4, simplify and reduce the wording in this column. This will make the follow up times easier to follow for the reader.

Response: Thank you for your comment. We have made the suggested change. 

Reviewer #3: This paper adequately summarises the growing number of investigations into COVID-19 related olfactory dysfunction, and provides succinct evidence to counsel patients on the expected recovery of olfactory dysfunction. The article is technically sound, and the data supports the conclusions made.

The methods were appropriate, and the PRISMA reporting guideline for systematic reviews was followed.

All supplementary data was made available for review

There are no concerns regarding the methods, reporting of results, or discussion.

Response: Thank you for your comments.

---

## [Editor Report · Decision Letter 1]

18 Oct 2021

Olfactory recovery following infection with COVID-19: A systematic review

PONE-D-21-25736R1

Dear Dr. Kilty,

We’re pleased to inform you that your manuscript has been judged scientifically suitable for publication and will be formally accepted for publication once it meets all outstanding technical requirements.

Kind regards,

Eng Ooi

Academic Editor

PLOS ONE

Additional Editor Comments (optional):

Thank you for making the changes. I've reviewed the revised manuscript and it addresses the reviewers comments
---

## [Editor Report · Acceptance letter]

25 Oct 2021

PONE-D-21-25736R1 

Olfactory recovery following infection with COVID-19: A systematic review 

Dear Dr. Kilty:

I'm pleased to inform you that your manuscript has been deemed suitable for publication in PLOS ONE. Congratulations! Your manuscript is now with our production department. 

Kind regards, 

on behalf of

Associate Professor Eng Ooi 

Academic Editor

PLOS ONE